# Effects of Selection on Breed Contribution in the Caballo de Deporte Español

**DOI:** 10.3390/ani12131635

**Published:** 2022-06-25

**Authors:** Ester Bartolomé, Mercedes Valera, Jesús Fernández, Silvia Teresa Rodríguez-Ramilo

**Affiliations:** 1Departamento de Agronomía, ETSIA, Universidad de Sevilla, 41013 Sevilla, Spain; mvalera@us.es; 2Departamento de Mejora Genética Animal, Instituto Nacional de Investigación y Tecnología Agraria y Alimentaria-Centro Superior de Investigaciones Científicas, 28040 Madrid, Spain; jmj@inia.csic.es; 3GenPhySE, Université de Toulouse, Institut National de Recherche pour l’Agriculture, l’Alimentation et l’Environnement, Ecole Nationale Vétérinaire de Toulouse, 31326 Castanet-Tolosan, France; silvia.rodriguez-ramilo@inrae.fr

**Keywords:** pedigree, open breed, breeding program, founders, show jumping, equine

## Abstract

**Simple Summary:**

The Caballo de Deporte Español (CDE) is a sport horse breed, which originated from crosses between different sport horse breeds in the search for a good sport aptitude for Dressage, Eventing and Show Jumping disciplines. The main aim of this study was to determine the effects of 15 years of selection on this breed and find out whether it has been effective and adequate regarding the CDE main breeding objectives. The whole known pedigree was used, comprising 47,884 animals (18,799 males and 29,085 females). Pedigree analyses were performed in order to check the population structure, origins and evolution over the years. For the analyses, animals were divided into fourteen breed groups. Performance data used in the analyses were the estimated breeding values (EBV) of the Show Jumping, Dressage and Eventing sport disciplines from the routine genetic evaluations of the CDE breeding programme. Results showed some degree of subdivision within the population and, therefore, inbred matings. Regarding the evolution of breeding values, we found that EBVs of offspring were higher than the EBVs of parents showing that genetic gain has actually occurred. Moreover, selection decisions seem to be taken by breeders based, to certain extent, on the genetic evaluations.

**Abstract:**

The equine breeding industry for sport’s performance has evolved into a fairly profitable economic activity. In particular, the Caballo de Deporte Español (CDE) is bred for different disciplines with a special focus on Show Jumping. The main aim of this study was to determine the effects of 15 years of selection and to find out whether it has been effective and adequate regarding the CDE main breeding objectives. The whole pedigree of 19,045 horses registered as CDE was used, comprising 47,884 animals (18,799 males and 29,085 females). An analysis performed to check for the pedigree completeness level yielded a number of equivalent complete generations (t) equal to 1.95, an average generation interval (GI) of 10.87 years, mean inbreeding coefficient (F) of 0.32%, an average relatedness coefficient (AR) of 0.09% and an effective population size (Ne) of 204. For the analyses, animals were divided into fourteen breed groups. Additionally, in order to study the evolution of these breeds over time and their influence on CDE pedigree, five different periods were considered according to the year of birth of the animals. Performance data used in the analyses were the estimated breeding values (EBV) of the Show Jumping sport discipline of 12,197 horses in the CDE pedigree, available from the 2020 routine genetic evaluations of the CDE breeding program (starting in 2004). Dressage and Eventing EBV values were also assessed. Results showed values of F higher than expected under random mating; this pointed to some degree of inbred matings. With regard to the evolution of breeding values, we found that, in general, EBVs of offspring were higher than the EBVs of parents. Notwithstanding, there is still a need for improvement in population management and the coordination of the breeders to get higher responses but controlling the loss of genetic diversity in the CDE breed.

## 1. Introduction

The equine breeding industry for sports performance has evolved in recent years into a fairly profitable economic activity. Thus, horse breeds have been selected to find the animal that is best suited for the equestrian discipline they participate in. This is true for pure breeds as well as for mixed breeds. In fact, different horse breeds, such as the Irish Sport Horse, Brazilian Sport Horse, Polish Sport Horse or, more recently, the Caballo de Deporte Español, were created as composite breeds, with the horses’ capacity for a certain sports performance as the objective or selection criterion [1]. In particular, the Caballo de Deporte Español (Spanish Sport Horse; CDE) is bred for Dressage, Eventing and Endurance disciplines with a special focus on Show Jumping discipline, taking advantage of the probable heterosis effect that results from this multiple breed combination [2,3]. The breed is managed by the Caballo de Deporte Español Breeders Association (ANCADES, www.ancades.com (accessed on 25 February 2022)), and its breeding program was officially approved in 2004 and updated in 2020 [4]. Its main goal is to select a horse with a suitable functional conformation, temperament and health, able to attain a high performance at either national or international sports events in which it participates. When the CDE Official Stud Book was created, an animal could be registered in the Foundational Registry (and thus be identified as a CDE animal) up until 2004 if it came from any of the “permitted” cross-breeds (established by Spanish laws APA/3318/2002 and APA/1646/2004) and was born between 1992 and 1998. Since the Foundational Registry was closed in 2004, an animal is considered to be a CDE either when both parents are registered as CDE, when only one of them is a CDE with the other parent from any of the “permitted” breeds included in these laws, or when both parents belong to any of these “permitted” breeds (even when both parents are from foreign breeds, their offspring could be inscribed as CDE at the owner’s request if they are not inscribed in any other official Stud Book). Although CDE is an open breed that allows animals of different breeds on its Stud Book [1], since the establishment of its breeding program, breeders are being encouraged to use animals already registered as CDE as reproducers. Notwithstanding, although heterosis falls in the F2 generation if purebred animals are no longer incorporated, an advantage in the phenotype of animals with a more “mixed” genetic background can still be manifested, with crosses being directed to obtain CDE horses showing a lot of heterosis [5]. However, no studies have been developed before about CDE breeders’ real preferences when selecting horses to create the next generation, i.e., whether they choose animals with the best estimated breeding values (EBV) for sport performance or just animals from the geographically nearest studs. Different criteria may have different repercussions on the outcome of the CDE breeding program. The selection process and mating are a consequence of a conscious decision, usually intended only for short-term achievements in intensively managed domesticated species [6]. In a crossbreed like the CDE horse, decisions are also influenced by breed preferences when choosing the individuals selected for breeding and, consequently, the evolution of performance and of the genetic parameters across time could also show this effect. The main aim of this study was to determine/characterize the effects of 15 years of selection on this breed and find out whether it has been effective and adequate regarding the main CDE breeding objectives.

## 2. Materials and Methods

### 2.1. Population Description

Since the creation of the CDE Official Stud Book (in 2002) until 2004, an animal could have been inscribed in the Foundational Registry (and thus be identified as a CDE animal) if it fulfilled two requirements: (i) coming from any of the admitted horse breeds (established by Spanish laws APA/3318/2002 and APA/1646/2004); and (ii) being born between 1992 and 1998. Since the closing of the Foundational Registry in 2004, not only animals with both parents registered as CDE can be included in the CDE Official Stud Book, but horses with only one or neither parent registered as CDE can also be included and considered CDE. In the latter cases, the “external” parents (i.e., non-CDE horses) should belong to any of the admitted breeds established in the CDE Stud Book. Obviously, their offspring can be inscribed in the CDE Stud Book if they are not already registered in any other official Stud Book. Furthermore, the non-CDE parents can be inscribed in the Auxiliary Registry of the CDE Stud Book as Foreign CDE (CDEx), and from that point, their descendants would be registered as CDE directly in the Principal Registry of the CDE Stud Book.

The pedigree information used in this study was gathered from the Caballo de Deporte Español official Stud Book, provided by ANCADES. For analysis purposes, no differences between CDE registrations were considered, thus counting 21,163 CDE horses for this study (19,045 registered as CDE in the Principal Registry and 2118 registered as Foreign CDE in the Auxiliary Registry of the Stud Book), representing 44.7% males and 55.3% females. The whole pedigree included 47,884 animals (18,799 males and 29,085 females).

According to the composite nature of this breed, different horse breeds were included in the CDE Stud Book via the relatives of the CDE animals (55.8% of the whole pedigree). These breeds were considered independently according to their representation in the CDE pedigree. Breeds with fewer than 500 animals (less than 1.5%) in the pedigree were grouped together in an “Other Horse Breeds” group (OHB). A total of 14 breed groups were assessed, 12 of them corresponding to majority horse breeds (>500 animals) present in the CDE Stud Book, together with the OHB and the CDE horses groups: Belgian Warmblood breed (BWP; 624 animals), Oldenburger breed (OLDBG; 642 animals), Westphalian breed (WESTF; 653 animals), Lusitano breed (LUS; 875 animals), Holsteiner breed (HOLST; 1269 animals), Anglo-Arab Horse breed (AAH; 1561 animals), Hanoverian breed (HANN; 1742 animals), Arab Horse breed (AH; 1835 animals), Dutch Warmblood (KWPN; 2120 animals), Selle Français (SF; 2311 animals), Thoroughbred (TH; 2924 animals), Other Horse Breeds group (OHB; 4731 animals), Pura Raza Española (PRE; 5434 animals) and the Caballo de Deporte Español breed group (CDE; 21163).

In order to study the evolution of these breeds over time (in terms of the proportion of individuals belonging to each breed) and their influence on CDE pedigree, five different periods were considered according to the year of birth of the animals. The first period (P0), including animals born before 2000, was settled before the Foundational Registry of the CDE Stud Book was created in 2002. Consequently, P0 only has animals from the Foundational Registry, and thus, no CDE animals were found in this period. Then, the second period (P1) included animals born between 2000 and 2004, the third period (P2), between 2005 and 2009, the fourth period (P3) involved individuals born between 2010 and 2014, and finally, the fifth and last period (P4), between 2015 and 2021. Figure 1 shows the distribution across these periods of the breed groups considered, showing different size and color of the circles according to the number of animals represented in each period. Furthermore, in order to confirm that all breeds considered for this study were genetically different between them, a principal component analysis was developed, using a matrix with the proportion of the individual genome coming from each original founder’s breed, calculated from the pedigree-derived relationship matrix (Figure A1).

### 2.2. Pedigree Analyses

Pedigree analyses were computed using the program ENDOG (v.4.8, [7]). In order to ascertain the circumstances affecting the genetic history of the CDE horse breed population, some parameters were calculated to determine the amount of available information, the pedigree completeness level, the number of complete generations (t) and the generation interval (GI) as a measure of the speed of genetic transmission due to the physiology of the species and the particular management performed. The first one was assessed as the proportion of ancestors known per generation for each offspring [8]. The t value was computed as the sum of (1/2)^n^, where n is the number of generations separating the individual from each known ancestor [9]. The GI was calculated as the average age of parents at the birth of their offspring kept for reproduction [10]. The three parameters were computed for the whole population and for a reference population, including only animals born in the last 10 years (between 2011 and 2021), which mostly corresponds to animals from the last generation. These will also be referred to as “active animals”.

In order to monitor the animal genetic resources for this breed, the effective population size (Ne) was also computed. PopRep 1.0. software’s (PopRep.tzv.fal.de (accessed on 15 June 2022)) [11] decision tree was used for picking the best method to calculate Ne for our data. After the analyses, the Ne calculated by individual increase in inbreeding, following Ref [12], appeared as the method that best fitted our data. Ne was calculated as $Ne=\frac{1}{2*\Delta F},\;where\;\Delta F=\frac{Ft-Ft-1}{1-Ft\;-\;1}$, where *F_t_* is the inbreeding coefficient of the offspring, and *F_t−1_* is the inbreeding coefficient of the parents.

### 2.3. Performance Data

The EBV of 12,197 horses for Show Jumping, 5253 horses for Eventing and 7790 horses for Dressage disciplines, from the CDE pedigree, were available for this study, including the performance data of animals from 2004 to 2020 [13].

Genetic parameters for the Show Jumping, Eventing and Dressage sport disciplines used in this study were obtained from the 2020 routine genetic evaluation information of the CDE breeding program. Genetic evaluation implies a multivariate BLUP analysis using the following genetic model:**y** = **Xb** + **Za** + **Wp** + **Qr** + **e**, 
where **y** was the vector of observations for the analyzed traits of each discipline; **b** was the vector of fixed effects; **a** was the vector of additive genetic effects; **p** the vector of rider-horse interaction effect (for Show Jumping discipline) or the rider effect (for Eventing and Dressage disciplines); **r** the vector of permanent environmental effects (for Show Jumping and Dressage disciplines); and **e** the vector of random residual terms, **X, Z, W** and **Q** were the incidence matrices assigning observations to the fixed, animal, rider-horse interaction or rider and permanent environmental effects, respectively.

In respect of the Show Jumping discipline, the analyzed traits were (i) penalty score transformed to a positive scale (ranging from 50 to 100, with 100 representing the highest punctuation in the competition “0 penalties” and 50 the lowest “maximum penalties”) and (ii) weighted total ranking (calculated on a positive points scale by assigning a value of 100 to the first classified animal (within the same event, penalty scale and competition level) and a value of 0 to the last) for Show Jumping. The fixed effects considered were: age, breed, gender, event, height of fences and category type of test interaction. The general index for this discipline combined both variables considered with a weight of 50% each.

For the Endurance discipline, the analyzed traits were (i) penalty score transformed to a positive scale of the Show Jumping exercise, (ii) final points obtained in the Dressage exercise and (iii) penalty score transformed to a positive scale of the Cross exercise. The fixed effects considered in this genetic model were: age, breed, gender, event, level of the event and owner, whereas the general index combined with a weight of 25% for (i), 35% for (ii) and 40% for (iii).

Finally, in the Dressage discipline, the analyzed traits were (i) points obtained at the dressage reprise, (ii) points for walk, (iii) points for trot, (iv) points for gallop, (v) points for submission and vi) points for general impression. This model considered the following fixed effects: age, breed, gender, event, level of the event and stud. Additionally, all traits were combined in the following general index: 70% for (i), 10% for (ii) and 10% for (iii), 5% for (iv), 2.5% for (v) and 2.5% for (vi).

It must be highlighted that, in all genetic evaluations, EBVs were standardized for an interval of 80–120 with a population average of 100. All BLUP genetic evaluations were carried out with the VCE software—version 6.0.2 (Nashville Video Production Company/Podcast Studio, Nashville, TN, USA) [14]. In addition, for the official genetic evaluations of the three disciplines considered, the accuracy of EBVs (r), was calculated as $r=\sqrt{1\;–\frac{PEV}{\delta_{a}^{2}}}$, where PEV is the prediction error variance and $\delta_{a}^{2}$ was the additive genetic variance.

### 2.4. Effect of Genetic Selection on Inbreeding and Average Relatedness Coefficient of the Caballo de Deporte Español Population

The level of genetic variability in the whole CDE population and within each breed group defined was assessed considering the following parameters:

Mean inbreeding coefficient (F), defined as the probability that an individual has two identical alleles by descent in any locus [15].

Average relatedness coefficient (AR) of each individual, interpreted as the representation of the animal in the whole pedigree, or the amount of the animal genetic information it shares. Thus, it is equivalent to the average relationship of an individual with the rest of the population, i.e., the marginal of the relationship matrix divided by the number of individuals. Furthermore, the average of all AR values is also equivalent to twice 1—the expected heterozygosity, which is a common measure of genetic diversity [16,17]. When AR is calculated for a founder, it expresses the proportion of the genetic information arising from it, and thus, its relevance in the development of the pedigree. In order to check the effect that breeders’ decisions have had when selecting the breeding stock on the genetic merit of the CDE population, the genetic variability was assessed for the whole population and also according to a hypothetical breeding scenario, where only CDE horses with CDE breed parents were allowed to register as CDE in the CDE official Stud Book.

Furthermore, using results from the Pedigree analyses developed before with the program ENDOG (v.4.8, [7]), the evolution across time of F and AR values for animals with best EBVs for the performance ability for Show Jumping, was also evaluated. The EBVs values used were described previously in Section 2.3.

### 2.5. Probability of Gene Origin of the CDE Horse Breed

Ancestors with both unknown parents in the available database were considered founders, as it is a common practice (unless molecular information exists for those animals).

In order to check the genetic representation of the founder individuals in the CDE population and in all the breed groups considered, the genetic contribution of founders to the descendant gene pool of the population [18] was assessed, classified by breed of origin and per period. This way the evolution of the relative influence of each breed in the composition of the CDE population could be studied. Additionally, for the 10 most contributing founders on each period (P0 to P4) their mean EBV for Show Jumping performance and that of their descendants on each period was computed to determine the genetic trend with time.

## 3. Results

### 3.1. Population Description

Breed groups’ distribution appearing in Figure 1 showed PRE and OHB as majority horse breeds in P0, with 4387 (19.2%) and 4076 (17.8%) animals born in this period (percentage within period), respectively. Afterwards, from P1 to P4, CDE was the majority horse breed, showing an increasing tendency from 79.4% (8804) animals in P1 to 98.2% (3986) in P4. On the other hand, PRE, OHB, KWPN, HANN, AH and SF breeds showed higher representation after CDE in P1, with 5.4% (596), 4.0% (442), 2.5% (276), 1.5% (169), 1.3% (144) and 1.1% (123) animals, respectively. All of them showed a decreasing tendency over the periods, with PRE still showing the highest representation in every period (after CDE), with 310 animals in P2 (4.9%), 121 in P3 (3.4%) and 20 in P4 (0.5%). The rest of the breeds showed fewer than 100 animals in P3 and P4 periods.

### 3.2. Pedigree Analyses

In order to study the quality of the pedigree information, for both the individuals included in the entire CDE Stud Book and those in the defined reference population (active animals), the pedigree completeness level was assessed. When the whole CDE population was considered, tracing back just two generations yielded a completeness level of 50%. The same level (50%) was obtained when analyzing the reference population, even when four generations were accounted for. Therefore, the quality of the pedigree information is higher for the reduced set of animals.

The number of animals, number of equivalent complete generations, average generation interval and effective population size, computed via individual increase in inbreeding for both the entire and the reference populations considered, were included in Table 1. The reference population represented 14.1% of the entire CDE pedigree and showed 1.91 more equivalent complete generations (3.86) and a generation interval 1.02 years longer (11.89) than the whole pedigree (with values of 1.95 and 10.87, respectively). On the other hand, Ne was 4% smaller in the reference population considered (Ne = 196) than in the whole pedigree (Ne = 204).

### 3.3. Effect of Genetic Selection on Inbreeding and Average Relatedness Coefficient of the Caballo de Deporte Español Population

The mean inbreeding coefficient (F) and average relatedness (AR) for the whole CDE population were 0.32% and 0.09%, respectively, and for the reference population, they were 0.79% and 0.19%, respectively, whereas the population with animals showing an EBV over the population mean (EBV > 100) showed 0.27% and 0.19% for F and AR values, respectively. The hypothetical scenario where only CDE horses were allowed to be registered and reproduced, showed an F coefficient of 2.18% and an AR of 0.19%.

The evolution of the above parameters in all the sets considered and of the Ne computed via individual increase in inbreeding over the five periods analyzed is shown in Figure 2. Considering that in an ideal population with random matings, F would be half the AR, the larger than expected *F*-values would indicate an important subdivision and/or non-random mating pattern in the population.

The results showed this subdivision in the three groups considered (total population, reference population and a hypothetical scenario with only CDE horses as reproducers), with a progressive increase over periods for the whole population, going from almost 0% F and AR values in P0 to 0.86% and 0.19%, respectively, in P4. For animals with EBV > 100, the *F*-values remained below 0.8% until P3, increasing considerably to almost 3.0% in P4, whereas the AR values remained below 0.4% over the four periods. Regarding the hypothetical scenario considered for the CDE population, the *F*-values increased considerably from almost 0.0% in P0 to almost 2.0% in P1 and P2 and continued increasing to 2.8% in P4. However, the AR values maintained a constant evolution with quite low values that went from 0.0% in P0 to a maximum of 0.25% in P4. In respect of Ne results, values in P0 and P4 were the lowest (around 180 animals), whereas P2 showed the highest Ne, with 237 horses.

### 3.4. Effects of Genetic Selection on the CDE Horse Breed

Genetic variances and heritability values from the genetic models used in the routine genetic evaluations of Show Jumping, Eventing and Dressage disciplines, are shown in Table 2. Heritability values were higher for Dressage than for Eventing or Show Jumping disciplines, ranging from 0.26 for Points obtained at the Dressage Reprise (P_R_), to 0.32 for Points for Gallop (P_G_) and Points for General Impression (P_GI_). On the other hand, Show Jumping discipline showed the highest variance values for all the genetic model components.

In order to ascertain the genetic change of the CDE breed regarding the objective trait, the EBV of Show Jumping, Eventing and Dressage performances characteristics and its evolution through the five periods considered was included in Table 3 for the most influential founders and their descendants.

Firstly, the 12 founders with the highest contributions to the entire CDE populations were included. Horses were ordered according to their year of birth. In the ‘Order’ column, the position of each founder in the ranking of global contributions and in each particular period was presented, showing the evolution of the ‘importance’ of each founder along time.

Results showed that the five most contributing founders for the whole CDE population were from TH (F201), OHB (F224 and F464) and SF (F333 and F445) breeds. They all showed an EBV for Show Jumping discipline above 101 (i.e., higher than the mean), and the mean EBVs of their descendants were between 101 and 102 (the superiority of these founders seems to be transmitted). In fact, there is a general upward trend through the periods for the EBVs of the descendants of the 12 founders listed, reaching up to 104. In respect of Eventing and Dressage disciplines, results showed that the ranking of individuals based on their EBVs was similar to the Show Jumping and Eventing performances but opposite to Dressage performance. These results are consistent with the negative genetic correlation found between Dressage and Eventing (−0.12 ± 0.0199).

When accounting for the evolution of the genetic contribution of founders and of their descendants’ EBVs over the P0 to P4 periods, most founders decreased their genetic contribution to the population across the years, together with their mean descendants’ EBVs, which remained below 100 from P0 to P4.

On the other hand, these founders showed mean accuracy values close to 40%, with maximum values above 70%.

The accumulated genetic contribution of all the founders from the CDE Stud Book, according to the breed group and the period, is represented in Figure 3, also including mean EBV for Show Jumping discipline for each breed’s founders and for their descendants. Breeds are ordered according to their breed founder’s accumulated genetic contribution. Only the results from the Show Jumping discipline were used for this analysis, as more data were available.

OHB and PRE founders showed the highest genetic contribution, at almost 15%, whereas LUS, WESTF, OLDB and BWP founders showed the lowest, with accumulated contributions of less than 5%. In contrast, the mean EBV of the descendants was considerably higher than their founders’ EBV for most breed groups, except for PRE, AH and LUS breed groups, in which we found the opposite outcome.

## 4. Discussion

Using composite breeds in domestic animals has been historically developed to benefit from the combinatory aptitude of this strategy in their progeny [19,20]. The amount of heterosis, as well as the particular outcome for any target trait, will depend on the proportion of genetic information coming from each of the breeds involved in the creation of the composite. With regard to the CDE population breed composition, preliminary analyses showed that the different breeds composing the CDE are still genetically different (there has been no homogenization of the breed). Notwithstanding, the genetic structure of the population has changed across generations, with CDE, PRE and OHB being the majority horse breeds in most periods. For CDE, the rising contributions (in percentage of the total number of horses of the CDE population) from P1 to P4 indicated a trend of breeders to prefer CDE stallions and/or mares as reproducers of the next generation. To fulfill the recommendation by Ref [3] of one or two breed crosses to account for optimal sport performance, another breed in addition to CDE should be used to obtain optimal results.

Regarding the available pedigree information, completeness level was slightly higher than that found in this breed in a study developed a few years after the Foundational Registry of the official Studbook was closed [1] and in a study from [21], comparing CDE results with other closed purebred populations. However, it was still less complete than other sport horse breeds such as the French Trotter [22] or the AH [23], where completeness values were above 50% as far back as the seventh generation while it reached this level in only the two most recent generations in CDE. This tendency was corroborated with GI and Ne values, being lower than those reported in the previous CDE paper [1]. These results also highlighted a better maintenance of the genetic diversity by year in this breed. These results, together with the lower t results than those reported in the previous CDE paper, could be due to foreign animals entering the studbook with some historic pedigree information (even if the amount of foreign horses was not large). Previous authors [24] reported similar values in the Brazilian Sport Horse.

When accounting for AR and F, the three sets of horses considered (whole CDE pedigree, animals with EBV > 100 and only CDE with CDE parents) showed greater F than expected under random mating from the corresponding AR values, thus indicating an important subdivision and/or non-random mating effect in these populations. A similar outcome was reported by Ref [1]. However, lower F and AR values has been observed in P4 (exclusive of the present study) than in the previous analyses. This could be due to the constant importation of foreign genetic material with limited information on the genealogy of the imported horses, so that, for F and AR calculations, these new foreign animals were considered as not inbred and not related to the rest of the studbook. Similar results were found in other sport horse breeds with open Stud Books [21,25]. Notwithstanding, of particular note is the large increase of mean *F*-value (mainly occurring in P4) for horses with EBV over the mean. This could indicate a trend in this last period to select CDE horses with higher EBV values according to Show Jumping performance genetic evaluations. These horses would tend to be more inbred by selecting parents coming from the same better performing ancestors. On the other hand, the simulation made of CDE with CDE parents’ population, showed increased *F*-values through all periods, indicating that genetic variability would be considerably affected within a few years if the Stud Book policies become more restrictive, as already reported in [1]. Considering Ne, despite values in all periods being in accordance with other open sport horse breeds [21,24], the decreasing trend from P2 to P4 could be due to a loss of genetic variability within the population, probably due to the selection efforts developed by breeders for Show Jumping, Eventing and/or Dressage performance, selecting only those animals that better fit their sporting purposes. Attention should be paid to this selection effect, in order not to lose critical genetic variability within this population in the future.

Results from the genetic contribution of the CDE founders, indicate that the five most contributing founders to CDE’s genetic diversity were derived from TH, OHB and SF breed groups, all with EBVs for Show Jumping performance over the mean and transmitting also high EBVs to their descendants throughout all periods. This observation makes sense considering these breeds share similar performance goals with CDE, with the Show Jumping discipline being the main breeding selection objective for most of them [26,27,28,29]. These results also suggest that CDE breeders have been making selection based on EBV’s results, as the higher the EBV of the founder (and of its descendants over periods), the higher genetic contribution it has had on the CDE population. This trend has been observed previously in other sport horse breeds [25,30,31,32,33]. On the other hand, we also found four founders from PRE and AH breed groups that, despite being within the list of the twelve most contributing founders, showed EBVs for Show Jumping below the mean and transmitting poor Show Jumping aptitude through all periods. However, when analyzing these founders for Eventing and Dressage results, they all show EBVs over the mean for both disciplines and good transmission to their progeny. This could be due to these founders being selected not for their genetic potential for Show Jumping performance but for Dressage, thus explaining the low EBV values for the former discipline. This different genetic selection was also supported by correlations found between disciplines, so that selecting for Dressage discipline would decrease performance in Eventing. Previous studies reported PRE as mainly bred for Dressage discipline [34,35], whereas AH were bred mainly for Endurance discipline [23], which shares some resilience and agility aptitudes with Eventing that are required for horses to compete successfully [36,37].

Furthermore, when comparing the heritability estimates for the three disciplines with previous studies, some differences were found. As regards to Show Jumping discipline, estimates were similar to those reported both in KWPN (0.11 [29]) and in a previous study for the CDE breed (0.047 to 0.085 [13]). When accounting for Eventing, heritability estimates in our study were higher than those reported in Great Britain’s sport horse breeds (0.05, [32]) and of lower magnitude than previous estimates in the CDE horse (0.16, [3]). For Dressage discipline, our heritability values were higher than those reported in KWPN (0.11, [29]), and in British native horse breeds, Arabs and Warmbloods (0.110 to 0.152 [33]), but in the range of those reported for PRE horse breed (0.22 to 0.59, [34]). Differences in heritability estimates could be due to the different methods used for the estimation and the populations sampled, explaining the lower differences found with previous studies in national breeds.

On the other hand, when accounting for the genetic contribution of all founders (not just the highest performing), it seems that breeders tended to choose minority horse breeds (OHB) and PRE horses to form their CDEs. As regards the OHB group, the reason could be this breed incorporates several foreign sport horse breeds that have been selected for sport performance, particularly Show Jumping, before the CDE breed was created [27,29,37,38]. Whereas, for PRE, this could be due to its position as the main native national horse breed in Spain and geographically distributed all around the country [34,39], thus making it a very handy option for most CDE breeders. However, as reported previously, PRE horses transmit a better Dressage genetic potential than other breeds, thus PRE founder’s breed groups EBV results were low, transmitting a poor genetic potential for Show Jumping to their descendants, as expected due to their main breeding orientation for Dressage.

Thus, it seems that a selection directed to different performance purposes has been made in CDE, with Eventing, Dressage and Show Jumping disciplines being selected at the same time. It has also to be noted that mean descendant’s EBV showed higher values than their founder’s for most breed groups, hence denoting genetic progress in the CDE breed. Therefore, selection has been effective in this population Although management has not been perfect, it seems that the selection of main breeding stock of the CDE breed was based on genetic results on sport performance of both selected stallions/mares and their descendants. Notwithstanding, despite descendants showing better EBVs than their CDE founders, there is still margin for improvement and the breeding association must make an effort to foster the performance of this breed by coordinating the management actions and encouraging the breeders to follow their recommendations for the selection and mating of the horses. Moreover, for future studies, if the levels of inbreeding increase, genetic models for sport performance in this breed could be improved by including pedigree inbreeding to account for inbreeding depression, as reported previously for other parameters [40]. However, for future studies, an introgression analysis using genotypic data would help reveal the real contributions of other breeds in the CDE pedigree.

## 5. Conclusions

Our results showed that, while a genetic progress can be observed in the CDE breed, with selection having an effect over this population, an improvement in population management is still needed to control the loss of genetic variability that could materialize if only CDE horses with high genetic value are used as reproducers, without implementing any control on the rate of increase in inbreeding. Thus, an effort should be made by ANCADES to persevere with the task of coordination and education of their breeders to make them aware of what genetic evaluations mean and how to use this breeding tool adequately. Furthermore, an effort should be made to complete the pedigree depths of all breeds connected by relatives of CDE animals in order to improve the accuracy of the breeding evaluations and, hence, enhance the genetic progress of this breed.

## Figures and Tables

**Figure 1 animals-12-01635-f001:**
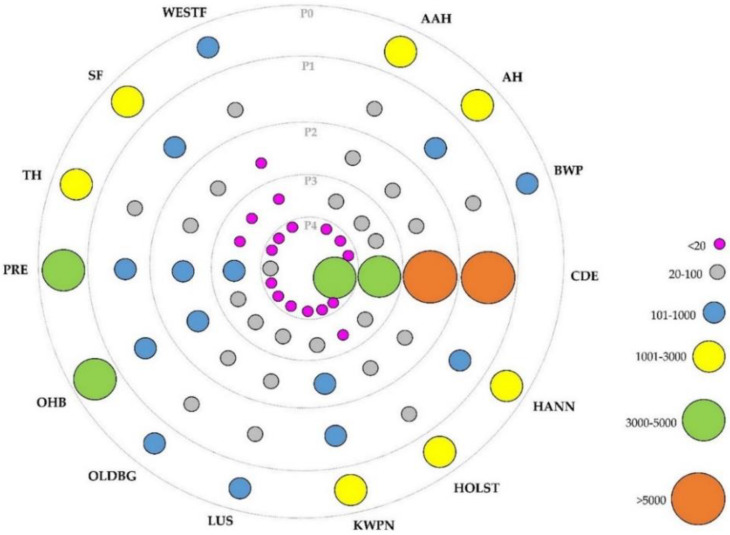
Distribution of horse breeds from the Caballo de Deporte Español Official Stud Book according to the period considered. Different sizes and colors of the circles indicate the number of animals of a certain breed within periods. Where AAH is Anglo-Arab Horse, AH is Arab Horse, BWP is Belgian Warmblood, CDE is Caballo de Deporte Español, HANN is Hanoverian Horse, HOLST is Holsteiner Horse, KWPN is KWPN Horse, LUS is Lusitano Purebred, OLDBG is Oldenburg Horse, OHB is Other Horse Breeds, PRE is Pura Raza Española, TH is Thoroughbred, SF is Selle Français, WESTF is Westphalian Horse; P0 is the period with animals born before 2000, P1 between 2000 and 2004, P2 between 2005 and 2009, P3 between 2010 and 2014, and P4 after 2015.

**Figure 2 animals-12-01635-f002:**
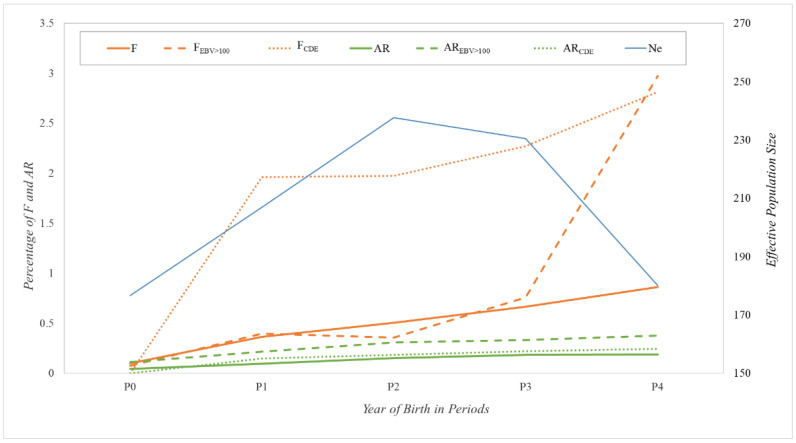
Evolution of average global inbreeding coefficient (F; orange solid line), for animals with breeding value over the mean for Show Jumping discipline (F_EBV > 100_; orange dashed line) and for animals with both parents from CDE (F_CDE_; orange dotted line). Evolution of average global relatedness coefficient (AR; green solid line), for animals with breeding value over the mean for *Show Jumping* discipline or (AR_EBV > 100_; green dashed line) and for animals with both parents from CDE (AR_CDE_; green dotted line). All values are in percentage. Evolution of effective population size computed via individual increase in inbreeding was also included (Ne; blue solid line). Where P0 is the period with animals born before 2000, P1 between 2000 and 2004, P2 between 2005 and 2009, P3 between 2010 and 2014, and P4 after 2015.

**Figure 3 animals-12-01635-f003:**
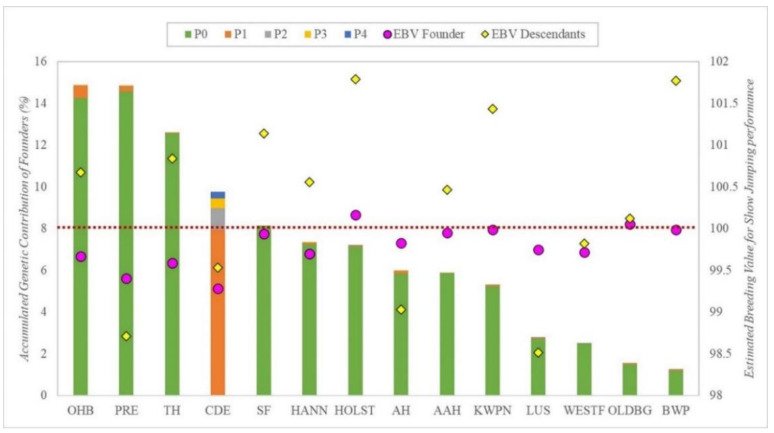
Genetic contribution of all founders classified by breed (each column), per period (different colors within columns), with their mean estimated breeding value (EBV Founder; pink circles) and the mean estimated breeding value of their descendants (EBV Descendants.; yellow rhombuses) for *Show Jumping* discipline, ordered according to contribution level of the founders. Where AAH is Anglo-Arab Horse, AH is Arab Horse, BWP is Belgian Warmblood, CDE is Caballo de Deporte Español, HANN is Hannoverian Horse, HOLST is Holsteiner Horse, KWPN is KWPN Horse, LUS is Lusitano Purebred, OLDBG is Oldenburg Horse, OHB is Other Horse Breeds, PRE is Pura Raza Española, TH is Thoroughbred, SF is Selle Français, WESTF is Westfalian Horse, P0 is period with animals born before 2000, P1 between 2000 and 2004, P2 between 2005 and 2009, P3 between 2010 and 2014, P4 after 2015. The red dots line indicate the mean estimated breeding value (EBV = 100).

**Table 1 animals-12-01635-t001:** Number of animals, number of equivalent complete generations (t), average generation interval (GI) and effective population size computed via individual increase in inbreeding (Ne) for the whole Caballo de Deporte Español Studbook and for the animals born between 2011 and 2021 (reference population).

Population	Number of Animals	*t*	GI	Ne
Whole Pedigree	47,884	1.95	10.87	204
Born between 2011 and 2021	6743	3.86	11.89	196

**Table 2 animals-12-01635-t002:** Estimation of variance components and heritability (h2) for the genetic evaluations computed for Show Jumping (SJ), Eventing (E) and Dressage (D) disciplines.

Variables		Variance Genetic Components						h^2^ (±SE)
		Animal	Rider	Rhi	Pe	Res	TV	
SJ	PS	93.22	-	193.43	55.58	1988.22	2330.44	0.04 (±0.0009)
	WTR	112.86	-	73.21	49.44	893.06	1128.57	0.10 (±0.0007)
E	PS_SJ_	8.70	1.26	-	-	33.63	43.60	0.12 (±0.2635)
	FP_D_	11.93	13.01	-	-	36.19	61.12	0.12 (±1.0982)
	PSc	41.83	30.04	-	-	281.05	352.91	0.12 (±2.9944)
D	P_R_	4.09	5.53	-	6.28	9.38	25.29	0.26 (±0.0574)
	P_W_	0.15	0.14	-	0.18	0.23	0.70	0.20 (±0.0002)
	P_T_	0.14	0.11	-	0.19	0.21	0.65	0.31 (±0.0085)
	P_G_	0.14	0.11	-	0.18	0.19	0.62	0.32 (±0.0011)
	P_S_	0.13	0.11	-	0.19	0.21	0.64	0.30 (±0.1520)
	P_GI_	0.14	0.11	-	0.19	0.19	0.62	0.32 (±0.0009)

Where PS = penalty score transformed to a positive scale; WTR = weighted total ranking; PS_SJ_ = penalty score transformed to a positive scale of the Show Jumping exercise; FP_D_ = final points obtained in the Dressage exercise; PS_C_ = penalty score transformed to a positive scale of the Cross exercise; P_R_ = points obtained at the dressage reprise; P_W_ = points for walk; P_T_ = points for trot; P_G_ = points for gallop; P_S_ = points for submission; P_GI_ = points for general impression; Animal = additive genetic effect; Rider = rider effect; Rhi = rider-horse interaction effect; Pe = permanent environment effect; Res = residual effect; TV = total variance; and h2 = heritability.

**Table 3 animals-12-01635-t003:** Evolution of the genetic contribution of the twelve most contributing founders of the Caballo de Deporte Español breed in period 0, ordered by their contribution for each of the periods P1 to P4 (in percentage), with their mean estimated breeding value (EBV) and the mean EBV of its descendants (EBV_d_) globally and per period, calculated based on the genetic evaluation of 2020 for *Show Jumping* (SJ), *Eventing* (Ev) and *Dressage* (Dr) disciplines.

Founders			F201	F464	F224	F445	F333	F203	F416	F1910	F229	F429	F430	F968
Year of Birth			1939	1950	1941	1950	1947	1939	1949	1963	1941	1949	1949	1956
Sex			M	F	M	F	F	F	M	M	M	M	F	M
Breed			TH	OHB	OHB	SF	SF	TH	TH	HOLST	AH	PRE	PRE	PRE
Contribution (%)			0.45	0.44	0.44	0.4	0.28	0.25	0.25	0.24	0.23	0.17	0.12	0.1
Order			1	2	3	4	5	6	7	8	9	10	11	12
Nd			5223	6641	6640	4632	3265	3672	3672	3233	1860	1623	1448	663
Global	EBV	SJ	101.3	101.5	101.5	101.7	102.4	103.2	103.2	98.9	98.6	98.7	99.3	99.9
		Ev	108.4	106.1	106.1	105.1	109.8	107.4	107.4	107.8	108.2	107.6	108.9	111.1
		Dr	98.7	97.2	97.5	96.8	100	99.4	99.4	100.7	100.4	107.2	102.7	-
	EBV_d_	SJ	101.3	101.2	101.2	101.4	101.1	102.3	102.3	102.5	97.3	98.9	99.2	98.8
		Ev	107.2	107.2	107.2	107.2	107.9	107.2	107.2	108.1	106.5	108.4	108.5	111
		Dr	99.9	98.2	98.2	98.2	100.2	99.9	99.9	100	99.1	103	103	97.7
P0	Order		4	1	2	3	17	8	7	45	6	5	10	9
	EBV_d_	SJ	101.9	101.3	101.3	101.5	101.6	102.4	102.4	101.9	98.9	98.9	98.9	99.2
		Ev	107.7	106.6	106.6	106.4	109	107.2	107.2	107.9	106.4	108.8	108.8	111.1
		Dr	99.2	97.6	97.6	97.5	99.6	98.4	98.4	99.8	99.6	103	103	97
P1	Order		3	1	2	4	5	28	27	51	6	39	58	74
	EBV_d_	SJ	100.2	100	100	100.4	100.2	101.3	101.3	101.8	96.1	99.2	99.3	96.3
		Ev	107.4	107.8	107.8	108	108.1	106.9	106.9	108.1	106.7	107	107.4	111
		Dr	98.5	97.6	97.6	97.6	99	99.4	99.4	99.8	97.1	103.7	103.7	98.2
P2	Order		1	2	3	4	6	8	7	5	16	27	55	90
	EBV_d_	SJ	101.7	101.9	101.9	101.9	101.3	102.7	102.7	102.6	92.7	98.3	99.8	97.9
		Ev	107.9	107.9	107.9	107.9	108.3	107.9	107.9	108	109.8	108.1	108.1	108.3
		Dr	99.7	99.5	99.5	99.3	99.8	100.4	100.4	99.3	97.3	100.6	100.5	101.8
P3	Order		1	3	4	5	12	10	9	2	22	56	73	103
	EBV_d_	SJ	102.3	102.6	102.6	102.4	102.5	102.6	102.6	102.9	95.1	99.6	100.4	99.9
		Ev	104.5	105.5	105.5	106.2	104.9	107	107	108.4	106.5	108	108	-
		Dr	104.5	104.9	104.9	104.7	104.5	104.2	104.2	101.6	97.5	104.6	104.6	103.6
P4	Order		1	3	4	2	18	16	17	5	109	101	129	461
	EBV_d_	SJ	102.2	102.9	102.9	102.7	103.7	103.3	103.3	104.9	94.3	-	-	-
		Ev	99	102.5	102.5	102.3	103.1	100.7	100.7	-	-	-	-	-
		Dr	104.3	103.2	103.2	103.2	104.5	100.5	100.5	102	-	106.3	106.3	104.7

Order indicates the order of the founder according to the percentage of contribution in the period/group considered, Nd is the number of all descendants of the founder throughout all periods, P0 is period with animals born before 2000, P1 between 2000 and 2004, P2 between 2005 and 2009, P3 between 2010 and 2014, and P4 after 2015.

## Data Availability

Restrictions apply to the availability of these data. Data were obtained from Asociación Nacional de Criadores de Caballo de Deporte Español (ANCADES) and are available from the corresponding author with the permission of ANCADES.

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
