# Peer review of "Effects of Selection on Breed Contribution in the Caballo de Deporte Español"

_animals, 2022, doi:10.3390/ani12131635_

Round 1

Reviewer 1 Report

Bartolomé et al. describe the effects of selection on breed contribution in the Caballo de Deporte Espanol. Contemporary population studies are using genomic data to explore the population structure of breeds, whilst in this study solely pedigree information was applied focusing on inbreeding and average genetic relationship. Furthermore the authors divided into fourteen breed groups. In this context, it would have been quite interesting to also investigate the relatedness of these groups and to show that they are genetically different, by applying for example a hierarchical clustering or PCA. I also do not understand that the authors have not calculated the Ne over time and only focused on the average inbreeding and relatedness. As aforementioned this study would have been much more interesting to read if authors also performed some genomic analysis. Furthermore I noticed some typos:

L21: inbred matings instead of mattings

L28: and to find out

L29: the whole pedigree about

L50: as well as for mixed breeds

L50: e.g. Irish sport horse

L59: select instead of obtain

L68: selecting horses

L94: registered instead of inscribed

L182: within each breed group

L192: register instead of inscribe

L251: registered

L348: Regarding the available pedigree information

L372: to select CDE horses

L374: by selecting parents with the same genetic origin

L380: were derived from

L40: inbred matings

L385: include EBV's results in the selection

Author Response

Reviewer 1

Comments to Author:

Bartolomé et al. describe the effects of selection on breed contribution in the Caballo de Deporte Espanol. Contemporary population studies are using genomic data to explore the population structure of breeds, whilst in this study solely pedigree information was applied focusing on inbreeding and average genetic relationship.

Furthermore the authors divided into fourteen breed groups. In this context, it would have been quite interesting to also investigate the relatedness of these groups and to show that they are genetically different, by applying for example a hierarchical clustering or PCA.

ANS: In our study, breed groups are classified due to officially recognized breeds. Thus, they already have recognized racial patterns that differentiates them and also makes them genetically different. We apologize but, as in this study we only use genealogical data, hierarchical clustering or PCA is not possible. However, we agree that it would be of great interest for future studies on this breed, to check molecular relations within these breeds.

I also do not understand that the authors have not calculated the Ne over time and only focused on the average inbreeding and relatedness.

ANS: Following your suggestions, Ne evolution was calculated and included in Figure 2, Ne for the whole population was included in Table 1 and some text explaining this parameter was included in Material and Methods (L163-168), Results (L272-273; L277-278; L280; L292-293; L304-305; L317-319), Discussion (L420-426) and Conclusions (L480) sections.

As aforementioned this study would have been much more interesting to read if authors also performed some genomic analysis.

ANS: We understand your concerning and agree that developing this study with genomic data would have been certainly more interesting. However, we can only use official information from the breed and, up to date, no genomic data was available for this breed. We will have your suggestion in mind for future studies.

Furthermore I noticed some typos:

L21: inbred matings instead of mattings

ANS: Word changed (L21)

L28: and to find out

ANS: Word included (L29)

L29: the whole pedigree about

ANS: Words changed (L30)

L50: as well as for mixed breeds

ANS: Word included (L51)

L50: e.g. Irish sport horse

ANS: Expression modified (L51)

L59: select instead of obtain

ANS: Word changed (L60)

L68: selecting horses

ANS: Word changed (L79)

L94: registered instead of inscribed

ANS: Word changed (L104)

L182: within each breed group

ANS: Word changed (L217)

L192: register instead of inscribe

ANS: Word changed (L232)

L251: registered

ANS: Word changed (L290)

L348: Regarding the available pedigree information

ANS: Sentence rephrased (L389)

L372: to select CDE horses

ANS: Sentence rephrased (L413-414)

L374: by selecting parents with the same genetic origin

ANS: Sentence rephrased (L415-416)

L380: were derived from

ANS: Sentence rephrased (L427)

L40: inbred matings

ANS: Word changed (L39)

L385: include EBV's results in the selection

ANS: Paragraph rephrased (L432-433)

Reviewer 2 Report

Your paper will be of interest to a specific audience - those concerned with the development and performance of the CDE.

Whilst having access to a large dataset I think restricting the tools you have used to evaluate it to only Endog ( and I have used Endog a lot myself) is a missed opportunity.

You cite other software by Groeneveld but have steered away from PopRep analysis. You clearly realise the implications of inbreeding upon the breed but PopRep would help paint a better picture of historic inbreeding as well as of Effective Population Size (Ne). You quite rightly say in your conclusions that awareness and management of inbreeding are significant for future breed management. Highlighting historic changes in Ne as well as current status would focus the minds of both Breed Society and individual breeders.

Whilst you report results based on Show Jumping EBV's in your Table 2 in the main body of the paper, you relegate the comparable results based on Eventing and Dressage to supplementary material.

This may be your own bias or from pre- assessment of results, but I believe the paper would be improved , less biased and more easily interpreted if you incorporate all three evaluations into one table in the main paper.

There are moderate changes to grammar needed and I have attached a marked up version of the pdf file to assist.

Author Response

Reviewer 2:

Comments to Author:

Your paper will be of interest to a specific audience – those concerned with the development and performance of the CDE. Whilst having access to a large dataset I think restricting the tools you have used to evaluate it to only Endog (and I have used Endog a lot myself) is a missed opportunity. You cite other software by Groeneveld but have steered away from PopRep analysis. You clearly realise the implications of inbreeding upon the breed but PopRep would help paint a better picture of historic inbreeding as well as of Effective Population Size (Ne). You quite rightly say in your conclusions that awareness and management of inbreeding are significant for future breed management. Highlighting historic changes in Ne as well as current status would focus the minds of both Breed Society and individual breeders.

ANS: In open populations like the CDE breed, historical inbreeding is difficult to be integrated into the new population unless good prior information is available. It would be necessary to recover previous genealogies of each individual included in the pedigree, otherwise, “founders” would be assumed to be not inbred and not related to other animals, which might not be true. In our population, an increase in mating between CDE animals was found only in recent years. Thus, only in case these mating increase a lot in the future, would comprise problems derived from inbreeding.

On the other hand, Ne was calculated with PopRep software and the Ne calculation method with best fit was included in the manuscript. The evolution of Ne over periods was included in Figure 2, Ne for the whole population was included in Table 1 and some text explaining this parameter was included in Material and Methods (L163-168), Results (L272-273; L277-278; L280; L292-293; L304-305; L317-319), Discussion (L420-426) and Conclusions (L480) sections.

Whilst you report results based on Show Jumping EBV's in your Table 2 in the main body of the paper, you relegate the comparable results based on Eventing and Dressage to supplementary material. This may be your own bias or from pre- assessment of results, but I believe the paper would be improved, less biased and more easily interpreted if you incorporate all three evaluations into one table in the main paper.

ANS: According to your suggestions, Table 2 was restructured and EBV’s for Eventing and Dressage were also included. Some more explaining was included in Material and Methods (L171-173; L196-207) Results (L334-337) and Discussion (L438-442) sections.

There are moderate changes to grammar needed and I have attached a marked up version of the pdf file to assist.

ANS: All changes were addressed and marked in red in the text.

Reviewer 3 Report

The main object of the manuscript (Effects of selection on breed contribution in the Caballo de Deporte Español) is an investigation of whether recent selection has been effective on genetic improvements of Caballo de Deporte Español in different sports disciplines including Show Jumping, Dressage and Eventing. The results of this study can help to design breeding programs for Caballo de Deporte Español. However, there are major and minor concerns that are summarized below:

Major concerns:

- I think that the introduction is too short. Please expand it. Also, describe history of  Caballo de Deporte Español breed in introduction.

Line 115-125: I am not sure if you use phantom pedigree (also called phantom parent groups). If you did not use it then I highly recommend using it in your analyses.

- You must define P0, P1, P2, P3 and P4 in caption of Figure 2. All figures must be standalone.

- You must define OHB, PRE and … in caption of Figure 3.

- What is the software you used for genetic evaluations?

- You must report variances you estimated and used in genetic evaluations.

- You must report genetic correlations between these traits.  

- I highly recommend reporting reliabilities (mean, max and min) of EBVs for different groups in this study.

- Line 63-164: “… transformed to a positive scale or the weighted total ranking q…”. In detail, you must describe how did you transform phenotypic records.

Minor concerns:

Line 59: Please change “good” to “suitable”.

Line 67-69: Please re-write from “However, no studies have been developed …” to “… animals from the, geographically, nearest studs.”

- I think a genetic introgression analysis using genotypic data can be helpful to reveal the contributions of other breeds in Caballo de Deporte Español breed. Therefore you can suggest it for future studies.

- Some parts of Table 2 and Table S1 are removed.

- Line 161: Please provide your model in a matrix form (for example y=Xb+Zu+Wp+e).

Author Response

Reviewer 3:

Comments to Author:

The main object of the manuscript (Effects of selection on breed contribution in the Caballo de Deporte Español) is an investigation of whether recent selection has been effective on genetic improvements of Caballo de Deporte Español in different sports disciplines including Show Jumping, Dressage and Eventing. The results of this study can help to design breeding programs for Caballo de Deporte Español. However, there are major and minor concerns that are summarized below:

Major concerns:

- I think that the introduction is too short. Please expand it. Also, describe history of Caballo de Deporte Español breed in introduction.

ANS: Following your suggestions, the introduction was expanded (L62-72; L78-81).

Line 115-125: I am not sure if you use phantom pedigree (also called phantom parent groups). If you did not use it then I highly recommend using it in your analyses.

ANS: In this study, founders are not usually related due to the open nature of this breed. Furthermore, most of these founder’s breeds are also open breeds, thus we did not consider this effect in this study due to its small weight both for the genetic evaluations and for the estimation of genetic variability. On the other hand, we have accounted for the genealogy above new animals entering each year in the CDE’s Studbook, as we completed up to the 3rd generation. However, we will consider your suggestion for further studies related with the EBVs in the CDE breeed.

- You must define P0, P1, P2, P3 and P4 in caption of Figure 2. All figures must stand alone.

ANS: Periods are defined in the caption of Figure 2 (L304-306).

- You must define OHB, PRE and … in caption of Figure 3.

ANS: Breeds are defined in caption of Figure 3 (L365-369)

- What is the software you used for genetic evaluations?

ANS: We used the VCE software – version 6.0.2. (L209-210)

- You must report variances you estimated and used in genetic evaluations.

ANS: Following your suggestions, variances were included in a supplementary Table S1, within Material and Methods Section, as these genetic evaluations were not developed specifically for this study. We included the variances used in the routine and official genetic evaluations used for each discipline (L213-214; L505-513).

- You must report genetic correlations between these traits.

ANS: Genetic correlations between the three genetic indexes estimated were included in the text (L337-338; L442-444).

- I highly recommend reporting reliabilities (mean, max and min) of EBVs for different groups in this study.

ANS: Following your suggestions, some results about reliabilities were included in Material and Methods (L210-213) and Results (L343-344) sections.

- Line 63-164: “… transformed to a positive scale or the weighted total ranking q…”. In detail, you must describe how did you transform phenotypic records.

ANS: According to your suggestions, some more explanations about the variables’ transformation was included in the text (L188-193).

Minor concerns:

Line 59: Please change “good” to “suitable”.

ANS: Word replaced (L60)

Line 67-69: Please re-write from “However, no studies have been developed …” to “… animals from the, geographically, nearest studs.”

ANS: Following your suggestion, part of this paragraph was changed for a better understanding (L78-81).

- I think a genetic introgression analysis using genotypic data can be helpful to reveal the contributions of other breeds in Caballo de Deporte Español breed. Therefore you can suggest it for future studies.

ANS: Following your suggestions, this recommendation was included within the text (L471-473).

- Some parts of Table 2 and Table S1 are removed.

ANS: This could be probably to their size. However, following also suggestions from other reviewers, Tables 2 and S1 were combined in one (Table 2). Thus, the structure was changed and the size was adapted to the journal’s margins. (Table 2: L345-353)

- Line 161: Please provide your model in a matrix form (for example y=Xb+Zu+Wp+e)

ANS: Genetic model was changed and text was adapted for the three disciplines (L176-207)

Round 2

Reviewer 1 Report

The authors addressed most of my comments. However, I do not a agree with the answer they provided to a major comment:

Furthermore the authors divided into fourteen breed groups. In this context, it would have been quite interesting to also investigate the relatedness of these groups and to show that they are genetically different, by applying for example a hierarchical clustering or PCA. ANS: In our study, breed groups are classified due to officially recognized breeds. Thus, they already have recognized racial patterns that differentiates them and also makes them genetically different. We apologize but, as in this study we only use genealogical data, hierarchical clustering or PCA is not possible. However, we agree that it would be of great interest for future studies on this breed, to check molecular relations within these breeds.

It is also possible to perfrom a PCA on a pedigree-derived relationship matrix (geneoalogical data) to investigate the population structure of your horses. For me this is quite important as the authors focus their analyses on different breed groups, without knowing if they are genetically differnt.

Author Response

The authors addressed most of my comments. However, I do not a agree with the answer they provided to a major comment:

Furthermore the authors divided into fourteen breed groups. In this context, it would have been quite interesting to also investigate the relatedness of these groups and to show that they are genetically different, by applying for example a hierarchical clustering or PCA. ANS: In our study, breed groups are classified due to officially recognized breeds. Thus, they already have recognized racial patterns that differentiates them and also makes them genetically different. We apologize but, as in this study we only use genealogical data, hierarchical clustering or PCA is not possible. However, we agree that it would be of great interest for future studies on this breed, to check molecular relations within these breeds.

It is also possible to perform a PCA on a pedigree-derived relationship matrix (genealogical data) to investigate the population structure of your horses. For me this is quite important as the authors focus their analyses on different breed groups, without knowing if they are genetically different.

ANS: Following your suggestions, we performed a PCA on genealogical data and figure was included in Appendix section (Figure S1). Some explanations were included within the manuscript (L135-139; L389-392). All breeds (or groups of breeds) appeared to be genetically different, as expected and, thus, performing the analyses considering this grouping is sensible.

Reviewer 2 Report

In its revised format the paper is much improved.

There are some minor revisions needed in use of English language prior to publication such as:

Line 29   The whole pedigree of 19,045 horses

Line 51   different horse breeds such as Irish Sport Horse,

Line 52   or more recently

Line 186 "In respect of the Showjumping" is better than "As regards to" - you have this occurring elsewhere in the manuscript as well.

Line 194  For the Endurance discipline

Line 200 Finally, in the Dressage discipline

Line 312   Delete "As Regards to" and start the sentence with "Ne values in P0 and P4 were the lowest"

Line 337/338   "EBV's that remained below 100"

Line 355      "used for this analysis as more data was available."

Line 368     "at almost 15%"

Line 415 0n    "Considering Ne, despite values in all periods being in accordance with other open sport horse breeds....."

Line 416     " the decreasing trend from P2 to P4"

Line 421    "critical"

Line 468    "would help reveal the real contributions of other breeds in CDE pedigrees."

Line 472   "improvement in population management"

Author Response

In its revised format the paper is much improved.

ANS: Thank you for your comments.

There are some minor revisions needed in use of English language prior to publication such as:

ANS: All changes were marked in red throughout the text.

Line 29   The whole pedigree of 19,045 horses

ANS: Word changed (L30)

Line 51   different horse breeds such as Irish Sport Horse,

ANS: Words changed (L51)

Line 52   or more recently

ANS: Word changed (L52)

Line 186 "In respect of the Showjumping" is better than "As regards to" - you have this occurring elsewhere in the manuscript as well.

ANS: Expression changed thoroughout the text (L189; L317; L335; L502)

Line 194  For the Endurance discipline

ANS: Words changed (L197)

Line 200 Finally, in the Dressage discipline

ANS: Word changed (L203)

Line 312   Delete "As Regards to" and start the sentence with "Ne values in P0 and P4 were the lowest"

ANS: Expression changed (L317)

Line 337/338   "EBV's that remained below 100"

ANS: Word changed (L343)

Line 355      "used for this analysis as more data was available."

ANS: Word changed (L350)

Line 368     "at almost 15%"

ANS: Word changed (L379)

Line 415 0n    "Considering Ne, despite values in all periods being in accordance with other open sport horse breeds....."

ANS: Sentence changed (L428-429)

Line 416     " the decreasing trend from P2 to P4"

ANS: Words changed (L430)

Line 421    "critical"

ANS: Word changed (L434)

Line 468    "would help reveal the real contributions of other breeds in CDE pedigrees."

ANS: Words changed (L481)

Line 472   "improvement in population management"

ANS: Word changed (L484)

Reviewer 3 Report

I think the manuscript is publishable.

Author Response

I think the manuscript is publishable.

ANS: Thank you for your comments.

Round 3

Reviewer 1 Report

The authors addressed all my comments

Author Response

The authors addressed all my comments.

ANS: Thank you so much for your comments.